# Leptospirosis-associated meningitis in an urban tropical endemic setting in northeastern Brazil: Three New cases and a meta-summary of 176 reported cases

Luís Arthur Brasil Gadelha Farias [1,2,3*], Osvaldo Mariano Viana Neto[4], Ednaldo Pereira Lima Sobrinho[4], Caroline Lucena de Almeida Vale[3], Geraldo Bezerra Silva Júnior[5], Elizabeth de Francesco Daher[4], Glaura Fernandes Teixeira de Alcântara[2], Antônio Silva Lima Neto[6], Tania Mara Silva Coelho[6], Maura Salaroli de Oliveira[1], Lauro Vieira Perdigão Neto[1,4]

**1** Department of Infectious Diseases of Hospital das Clínicas and Laboratório de Investigação Médica – LIM 49, University of São Paulo, São Paulo, Brazil, **2** São José Hospital for Infectious Diseases, Fortaleza, Brazil, **3** Christus University Center (Unichristus), Fortaleza, Brazil, **4** Department of Community Health, Faculty of Medicine, Federal University of Ceará, Fortaleza, Brazil, **5** University of Fortaleza (UNIFOR), Fortaleza, Brazil, **6** Health Department of the State of Ceará (SESA-CE), Fortaleza, Brazil

* luisarthurbrasilk@hotmail.com

## Abstract

### Background

Leptospirosis is a globally distributed zoonotic disease with a broad clinical spectrum. Central nervous system (CNS) involvement is uncommon and under-recognized, and leptospiral-associated meningitis (LAM) is primarily described in isolated case reports and small series. No study to date has integrated institutional cases with published reports to characterize cerebrospinal fluid (CSF) profiles in LAM.

### Methods

We conducted a retrospective cohort study of patients with meningitis, encephalitis, or meningoencephalitis admitted to a tertiary infectious diseases center in northeastern Brazil. Cases of LAM were identified among patients with aseptic meningitis. In parallel, a narrative literature review was performed, and a meta-summary of published cases was constructed. CSF parameters from both datasets were extracted and analyzed using descriptive statistics and graphical (boxplot) methods.

### Results

Among 809 patients with meningitis or encephalitis, 447 presented with aseptic meningitis. Three cases (0.67%) of LAM were identified, presenting as isolated meningitis or meningoencephalitis. Among 179 patients, including 176 identified in the medical literature, mean CSF values were: cellularity 68 cells/mm$^3$ (range, 30–7800),

**Data availability statement:** All relevant data are in the manuscript.

**Funding:** GBdS Jr is a recipient of research productivity funding from the Conselho Nacional de Desenvolvimento Científico e Tecnológico (CNPq), Brazil. The research scholarships provided by the Programa Institucional de Bolsas de Iniciação Científica (PIBIC) at the Federal University of Ceará (UFC) to OMVN and EPLS were directly related to and supported the development of this study. The funders had no role in study design, data collection and analysis, decision to publish, or preparation of the manuscript.

**Competing interests:** The authors have declared that no competing interests exist.

lymphocytes 73% (0–100), neutrophils 18% (0–96), glucose 60 mg/dL (0–140), and protein 67–90 mg/dL (31–2590). CSF findings in the institutional cohort showed mild to moderate pleocytosis with lymphocytic predominance, normal to elevated glucose levels, and increased protein concentrations. The integrated analysis of cohort and published cases—the first combined CSF profile synthesis of LAM—demonstrated a consistent pattern of lymphocytic pleocytosis, mildly elevated cellularity, preserved glucose levels, and increased protein, with substantial inter-case variability.

## Conclusion

LAM is an uncommon but clinically relevant cause of CNS infection in endemic settings. This study provides the first integrated synthesis of CSF profiles from both institutional cases and published literature, supporting a characteristic but variable CSF pattern in leptospiral CNS disease. Clinicians should consider leptospirosis in patients with aseptic meningitis in endemic areas, particularly when epidemiological risk factors are present. Improved diagnostic capacity and prospective studies using standardized criteria are needed to better define disease burden and refine diagnostic and therapeutic approaches.

---

### Author summary

In this study, we describe leptospirosis as an underrecognized cause of meningitis by reporting three cases from an endemic region in northeastern Brazil and combining them with data from 176 previously published patients worldwide. We show that this condition often resembles other forms of aseptic meningitis, making diagnosis difficult in routine clinical practice. By integrating new and existing data, we outline a general pattern of spinal fluid findings, typically showing mild inflammation with predominance of certain immune cells, while also demonstrating that results can vary widely between patients. This variability helps explain why the disease is frequently overlooked. Our findings suggest that leptospirosis associated meningitis is likely more common than currently reported, especially in settings where access to more sensitive diagnostic tools, such as molecular tests, is limited. Increasing awareness of this condition is important, as early recognition can guide appropriate treatment and improve patient outcomes. Overall, this work highlights a neglected presentation of a widespread tropical disease and provides practical information to support clinicians in endemic areas when evaluating patients with suspected meningitis.

## Introduction

Leptospirosis is a globally prevalent zoonotic disease caused by *Leptospira* spp., Gram-negative spirochetes that typically use mammals—mainly rodents—as their

natural reservoirs [1]. In tropical continental countries such as Brazil, leptospirosis can exhibit high incidence rates, particularly in endemic areas during the rainy season and following flooding events. In the Northeast region, between 2019 and 2022, 2,199 cases of leptospirosis were reported, with a case fatality rate of 13.8% [2]. The bacteria colonize the renal tubules of reservoir hosts and are excreted in their urine, contaminating the environment and enabling transmission to humans, who are considered incidental hosts [1]. The incubation period in humans is variable, ranging from one day to four weeks after exposure, and in survivors, the infection may persist for several months [3].

After contamination, the spirochetes disseminate through the bloodstream during an approximately eight-day period known as the leptospiremic phase, in which a higher pathogen load may result in more severe disease [1]. The subsequent immune phase corresponds to the immunological response to infection [4]. Clinically, leptospirosis can range from a mild febrile illness to a severe icteric–hemorrhagic syndrome with multiple organ dysfunction [5], first described by Adolph Weil in 1886 [6]. Despite extensive research, important gaps remain in understanding the pathogenesis of this tropical disease, particularly regarding the extent to which clinical manifestations result from direct bacterial invasion versus host immune-mediated injury [1].

Although it is described and frequently cited as part of the disease spectrum [7], leptospirosis-associated meningoencephalitis (LAM) remains poorly characterized in the medical literature. Most publications addressing this topic were released before the 21st century, often in low-impact scientific journals, and many of these reports are not available online, hindering the consolidation of current knowledge on LAM. Several factors may contribute to this gap: (i) LAM can clinically mimic other causes of aseptic meningoencephalitis, such as viral, mycobacterial or rickettsial infections [8]; (ii) appropriate diagnostic tools for central nervous system (CNS) involvement are often unavailable in regions where leptospirosis is endemic; and (iii) the empirical use of antibiotics during the course of meningoencephalitis may reduce the likelihood of isolating the pathogen [9].

Consequently, establishing a consistent clinical–epidemiological–laboratory profile of LAM remains challenging, as it may encompass a wide range of neurological syndromes, including meningitis [10], meningoencephalitis [9], acute encephalitis [11], demyelinating diseases [12,13], cerebrovascular disorders [14], focal CNS lesions [15], peripheral neuropathies [16], and Guillain–Barré syndrome [17], among others. In addition, important inconsistencies persist across current studies, largely driven by methodological limitations, including small sample sizes, reliance on isolated case reports, heterogeneous diagnostic criteria, and variability in laboratory methods, which collectively hinder meaningful comparisons and synthesis of findings. For example, some reports describe LAM exclusively in anicteric leptospirosis, whereas others demonstrate its occurrence in severe forms such as Weil's disease [18,19].

We characterized LAM cases among patients with aseptic meningitis at a tertiary infectious diseases center in Ceará, Northeastern Brazil, and performed a literature review and meta-summary of case reports, series, and cohorts to better define its clinical and cerebrospinal fluid (CSF) profiles.

## Methods

### Ethics statement

This study is part of a cohort approved by the Research Ethics Committee of the Hospital Sao Jose de Doencas Infecciosas (HSJ) (protocol Nº CAAE 52811521.7.0000.5044).

### Asseptic meningitis cohort and leptospiral-associated meningitis cases

This manuscript presents a retrospective observational study describing LAM cases, complemented by a structured literature review and a meta-summary of published cases.

This study comprised a comprehensive review of medical records from patients managed at a statewide referral center for infectious diseases between january 2017 and december 2024, which receives virtually all cases of meningitis and encephalitis in Ceará, a state in the Northeast region of Brazil with a population of approximately nine million inhabitants.

## Definitions

Acute encephalitis syndrome and meningitis were defined according to the 2003 WHO guidelines [20]. Acute encephalitis syndrome was characterized by sudden onset of fever with altered mental status (such as confusion, disorientation, coma, or aphasia) and recent seizures (excluding simple febrile seizures). Meningitis was defined as sudden fever (>38.5 °C rectal or 38.0 °C axillary) accompanied by neck stiffness, altered consciousness, or other meningeal signs. Patients meeting criteria for both conditions were classified as meningoencephalitis [8,20].

Aseptic meningitis/encephalitis/meningoencephalitis are commonly defined as a syndrome characterized by acute onset of signs and symptoms of meningeal inflammation, CSF pleocytosis and the absence of microorganisms on Gram stain and/or on routine culture [21].

A leptospirosis case was defined as a patient presenting with clinical features compatible with leptospirosis (e.g., acute febrile illness with headache, myalgia, conjunctival suffusion, or jaundice), with a relevant epidemiological exposure (such as contact with contaminated water or soil, occupational or recreational exposure, or residence in/travel to endemic areas), and laboratory confirmation by serology (IgM ELISA or microscopic agglutination test), PCR, or culture. LAM were defined as a leptospirosis confirmed case with the presence of compatible meningeal/encephalical syndrome [9,22].

## Literature review

A narrative review of the literature was conducted to identify cases of meningitis associated with leptospirosis. Searches were performed in electronic databases, including PubMed, Scopus, and Web of Science, using the terms "meningitis" AND "leptospirosis", covering studies published up to November 2025.. Duplicates were removed, and articles were independently screened by two authors (LABGF and OMVN). Studies reporting central nervous system involvement in leptospirosis were included, with emphasis on case reports, case series, and observational studies describing clinical, laboratory, or pathological features of LAM. Both molecular and serological diagnostic approaches were considered. Only articles published in English with sufficient clinical detail were included.

Studies describing cases of leptospirosis meningeal syndromes that met the predefined criteria were grouped, as shown in Table 2. Those providing defined CSF parameters were compiled into a separate dataset, from which the following variables were extracted whenever available: cell count (cells/mm$^3$), lymphocytes (%), neutrophils (%), CSF protein (mg/dL), and CSF glucose (mg/dL). When a study did not report a specific parameter, or when values were presented in non-compatible units (e.g., blood/CSF ratios), that parameter was excluded from the analysis for that study.

For each variable, the corresponding boxplot statistics were obtained (Q1, median [Q2], Q3, lower whisker, and upper whisker) and plotted together with the values from the three cases reported in the present study using Python (Matplotlib/Seaborn). To ensure uniformity and avoid distortion of the final boxplot distributions, extreme CSF values were excluded according to predefined biological plausibility thresholds. Cases with cellularity >1300 cells/mm$^3$ or protein levels >500 mg/dL were excluded, as they represented extreme outliers relative to the observed data distribution. No predefined biological range was assumed; however, a few markedly discrepant values substantially distorted the boxplot and influenced the visualization of central tendency and dispersion. Additionally, entries lacking all CSF quantitative variables were omitted to prevent incomplete cases from biasing the graphical analysis.

Missing data were visualized using a Seaborn heatmap. To ensure that the heatmap reflected the contribution of each study proportionally, the dataset was expanded so that each row represented an individual patient; for series-based studies, each record was replicated according to the reported number of patients.

## Results

### Aseptic meningitis cohort

Initially, 809 patients with confirmed meningitis/encephalitis/meningoencephalitis were identified. Among them, there were 447 patients with aseptic meningitis. From them, those with negative cultures and RT-PCR FilmArray meningitis/

encephalitis (ME) panel (BioFire Diagnostics, LLC, Salt Lake City, UT, USA), and a clinical suggestive picture underwent *Leptospira* serology. All patients with positive *Leptospira* IgM ELISA and confirmed meningeal syndrome recorded in their medical records were included in our analysis. Of the 447 patients, three patients (0.67%) confirmed leptospiral infection. Laboratory and medical data, including those at admission, were available for the two patients included in the analysis.

## Cases description

**Case 1**. A 22-year-old Brazilian man who works in an urban area as a motorcycle delivery worker, and without previous comorbidities, arrived at the emergency room (ER) with a 7-day history of high fever (Temperature 40°C), headache, nuchal stiffness, polyarthralgia, macroscopic hematuria and nausea. He reported prolonged occupational exposure (>1 hour) to stagnant rainwater one week prior to admission.

Vital signs included a heart rate of 94 bpm, a respiratory rate of 20 rpm, and a peripheral oxygen saturation (SpO$_2$) of 99% without supplementary oxygen. On physical examination, he presented with mild jaundice and a pain-related facial expression. Neurological examination revealed positive Kernig's and Brudzinski's signs, with a negative Lasègue sign. He was alert and oriented (Glasgow Coma Scale score of 15). Laboratory tests showed hyperbilirubinemia, thrombocytopenia, and renal dysfunction. Cranial CT was unremarkable. CSF analysis was consistent with aseptic meningitis (Table 1). Serum anti-Leptospira IgM ELISA (Panbio, Abbott, Australia) was positive, and microscopic agglutination testing (MAT) showed a titer of 1:400. The serovar was not identified.

Empirical treatment for bacterial meningitis with intravenous ceftriaxone (2 g, twice daily), vancomycin (15 mg/kg three times daily), and intravenous dexamethasone (12 mg, three times daily) was initially performed; vancomycin and dexamethasone were discontinued after leptospirosis confirmation. The patient improved clinically, although he developed acute pancreatitis during hospitalization, managed conservatively. He achieved full recovery and was discharged after four weeks.

**Case 2.** A 66-year-old Brazilian man, a construction worker with hypertension and alcohol use, presented with a 2-week history of fever, headache, polyarthralgia, disorientation, and sensory deficits. He reported a recent lower-limb laceration sustained during flood exposure.

Vital signs included a heart rate of 124 bpm, a temperature of 37,8ºC, a respiratory rate of 16 rpm, and a peripheral oxygen saturation (SpO$_2$) of 88% without supplementary oxygen. On physical examination, Glasgow Coma Scale score of 11, with marked sensory loss. Neurological examination revealed nuchal stiffness. Laboratory tests showed anemia, leukocytosis, elevated inflammatory markers, hyperbilirubinemia, and mild renal dysfunction. CSF findings were consistent with aseptic meningitis (Table 1). Serum anti-Leptospira IgM ELISA was positive (Panbio, Abbott, Australia). MAT was unavailable at the time. returned a positive result.

The patient received ceftriaxone and vancomycin and required ICU admission with mechanical ventilation due to neurological impairment. He improved clinically, with recovery of renal function and no need for dialysis. Other complications,

**Table 1. Cerebrospinal fluid profile of patients with severe leptospirosis-associated meningitis.**

|  | Cellularity (cells/mm³) | Differential (Linf/Mono/Neu/Eos) [%] | Glucose (mg/dL) | Protein (mg/dL) | ADA (IU/L) | Cultures and bacterioscopy* | PCR** |
|---|---|---|---|---|---|---|---|
| Case 1 | 33 | 75/10/14/01 | 76,0 | 31,9 | 2,2 | – | – |
| Case 2 | 18 | 81/05/14/00 | 132,0 | 93,0 | 4,9 | – | – |
| Case 3 | 9 | 85/05/10/00 | 64,8 | 54,4 | — | – | – |

"-": Negative results. *Cultures for mycobacteria, fungi, and pyogenic pathogens. **FilmArray meningitis/encephalitis (ME) panel (BioFire Diagnostics, LLC, Salt Lake City, UT, USA).

such as alveolar hemorrhage, were investigated through thoracic CT and ruled out. After 14 days of antimicrobial therapy, he was discharged without neurological sequelae, confirmed at 6-month follow-up.

Case 3. A 25-year-old transgender woman, a waste picker with newly diagnosed type 2 diabetes, presented with myalgia, calf pain, neck stiffness, and a recent convulsive episode. She reported occupational exposure to stagnant rainwater.

On admission, she was hemodynamically stable (temperature 37.8°C) and fully alert (GCS 15), with no focal neurological deficits. Laboratory evaluation revealed severe rhabdomyolysis (CPK 72,000 U/L) and acute kidney injury (creatinine 3.5 mg/dL, urea 115 mg/dL). Cranial CT was normal. CSF analysis demonstrated lymphocytic meningoencephalitis with low cellularity and protein (Table 1). Serum anti-*Leptospira* IgM ELISA (Panbio, Abbott, Australia) was positive.

She was treated with intravenous ceftriaxone and supportive care. Renal dysfunction was managed conservatively without dialysis. The patient remained neurologically stable, with no recurrence of seizures. She was discharged with partial renal recovery and outpatient follow-up.

### Literature review

In this study, we present three rare cases of a common disease - leptospirosis. Given the uncertainties regarding the possible axial and extra-axial manifestations of the disease, we conducted a literature review with a meta-summary of published case reports. Through this approach, we identified 44 articles describing 179 cases with available clinical and CSF profiles.

The retrieved articles were summarized in Table 2. The aggregated data show that, among the studies with available information on jaundice, the majority of patients with leptospiral meningitis/meningoencephalitis were anicteric (59.1%), while 40.9% presented with jaundice. Four studies did not express the prevalence of icteric/anicteric patients in their cohort [8,23–25].

The articles that provided CSF data were allocated to a spreadsheet; total missing data was 44.1% (Fig 1). A total of 179 patients had their CSF analyzed. A lymphocytic inflammatory pattern was described, with a median low-to-moderate cellularity (approximately 68 cells/mm$^3$) and wide variation, including a few very high values (Fig 2). The lymphocyte ratio is high (median close to 73%), while neutrophils remain low (median close to 18%). However, among the 42 patients for whom CSF neutrophil data were available, 12 (28.6%) demonstrated a predominance of neutrophils in the CSF. CSF glucose tends to be within the normal range (median ~60 mg/dL), and protein shows a slight to moderate elevation (median ~67–90 mg/dL), although with rare very high values. Taken together, the findings reinforce a typical profile of aseptic meningitis, with lymphocytic predominance, absent hypoglycorrhachia, and variable proteinorrhachia.

Two reports were excluded from Fig 2 due to markedly elevated protein levels in the CSF, measuring 875 mg/dL and 2590 mg/dL, respectively [15,39]. Additionally, two cases exceeded the predefined upper threshold for CSF cellularity, with 7800 cells/mm$^3$ and 5000 cells/mm$^3$, and were therefore also removed from the analysis [39].

## Discussion

The present study comprises a retrospective cohort of aseptic meningitis cases managed at a tertiary infectious diseases center in northeastern Brazil, with identification and characterization of confirmed cases of LAM.

In our cohort, 3/447 (0.67%) of patients with aseptic meningitis had serological evidence compatible with leptospirosis, including one case of isolated meningitis and two cases of meningoencephalitis, all of which exhibited lymphocytic predominance in the CSF. In contrast, Romero et al. (2010) [40] reported a 58.97% positivity rate for *Leptospira* spp. using molecular methods in CSF samples from patients with aseptic meningitis, a proportion substantially higher than that observed in our study. This discrepancy may be explained by methodological differences, as PCR-based techniques tend to demonstrate greater sensitivity [8]. In another study conducted in the same region, IgM ELISA positivity in CSF was observed in 14.6% of cases of meningitis of indeterminate etiology [43]. In a large series of leptospirosis, meningeal involvement has been reported in up to 37% of patients, and CSF pleocytosis in up to 47% [56]. Therefore, the likelihood that test positivity in the context of meningitis merely reflects incidental exposure to *Leptospira* is considered low [9].

**Table 2. Studies and cases on leptospirosis-associated meningitis in medical literature.**

| Study | Age | Country | Type | Icteric/ anicteric LAM's | CSF findings | Diagnosis | Interval between fever and meningeal syndrome | Findings |
|---|---|---|---|---|---|---|---|---|
| Nabity et al. (2020) [22] | 30.8 | Brazil | Retrospective cohort | 0/5 | ⅗ with monomorphic CSF pattern. | MAT + and clinical LAM diagnosis | – | Lower mean CSF cell count and protein (p < 0.05). 1,7% prevalence among aseptic meningitis patients. |
| Nordholm et al. (2019) [26] | 27 | Denmark | Case report | 0/1 | Neutrophilic pattern (56%) | MAT+ and clinical LAM diagnosis | 15 days | Report of pet mice leptospirosis. |
| Wilson et al. (2014) [27] | 14 | United States of America | Case report | 0/1 | Lymphocytic pattern (52%). Elevated ACE in CSF. | CSF NGS | 0 days | Chronic neuroleptospirosis in a patient with adenosine deaminase deficiency. MRI with hyperintensities in basal ganglia + leptomeningitis. |
| Wang et al. (2016) [28] | 37 | Taiwan | Case report | 0/1 | Lymphocytic pleocytosis (91%), 24 cmH$_2$O of opening pressure | MAT + clinical LAM diagnosis | 2 days | Absence of kidney injury, disease |
| Díaz-Rivera et al. (1958) [29] - icteric | – | Puerto Rico | Prospective cohort | 9/0 | Lymphocytic pleocytosis (mean value: 85,7%) | Serology | 8,1 days (mean) | 11,1% of cases with leptospiral CSF isolation. Three punctures with yellow collor. |
| Díaz-Rivera et al. (1958) [29] - anicteric | – | Puerto Rico | Prospective cohort | 0/7 | Lymphocytic pattern (mean value: 62,0%) | Serology | 9,3 days (mean) | 14,3% of leptospiral CSF isolation. |
| Bismaya et al. (2022) [30] | 30.7 | India | Retrospective cohort | 0/7 | Lymphocytic pattern in 100% of patients | Serology and PCR | – | 14,0 cmH$_2$O of mean opening pressure. 2 with an abnormal MRI. |
| Gala et al. [31] (2024) | – | Belgic | Case report | 1/0 | Lymphocytic pattern. | PCR | 9 days | MRI exhibiting subarachnoid exsudates, with smooth leptomeningeal enhancement in and communicating hydrocephalus. Xanthochromic PCR. Patient died. |
| Samkar et al. [32] (2015) | 28.5 | Netherlands | Case series | 0/1 | 2 Lymphocytic, 2 neutrophilic (mean: 80,5%) pattern | MAT/Serology/PCR/ Culture | – | In the review, a significantly increased mortality were found in patients not treated with antibiotics, when compared to those treated (13% vs 2%; p = 0.04). |
| Bandara et al. [33] (2021) | 31,5 | Sri Lanka | Case series | 0/2 | 1 Lymphocytic, 1 neutrophilic pattern | MAT | 7,5 days (mean) | All alive. |
| Souza et al. [34] (2006) | 19 months | Brazil | Case report | 0/1 | Lymphocytic pleocytosis (59%) | MAT | | Pediatric case. Normal CT scan. Long-term evolution favourable. |
| Mathew et al. [35] (2006) | 36.4 | India | Mixed cohort | – | Lymphocytic pleocytosis (72%) | MAT | | Normal CT scan on 18/27, 7/27 with diffuse cerebral oedema. 5/22 of CSF MAT samples were positive. 26% died. The two significant parameters for mortality were elevated CSF protein and the degree of altered sensorium at admission. Higher protein level in patients who died (p < 0.001) |
| Alani et al. [36] (1993) | 20 | United Kingdom | Case report | 1/0 | Neutrophilic pleocytosis (95%). | Serology/MAT | – | – |

*(Continued)*

| Study | Age | Country | Type | Icteric/anicteric LAM's | CSF findings | Diagnosis | Interval between fever and meningeal syndrome | Findings |
|-------|-----|---------|------|------------------------|--------------|-----------|----------------------------------------------|----------|
| Karande et al. [37] (2005) | 10 | India | Case report | 0/1 | Lymphocytic pleocytosis (94%). | Serology/MAT | 5 days | CT scan revealed inflammation along the tentorium, the ambient cisterns and the left sylvian fissure. |
| Tattevin et al. [38] (2003) | 29 | France | Case report | 0/1 | Neutrophilic pleocytosis (90%). | MAT | – | The meningeal syndrome started four hours after the administration of amoxicillin, and disappeared two days later. |
| Jha et al. [9] (2010) | 23 | India | Case Report | 1/0 | Lymphocytic pleocytosis (90%). | Serology | – | Normal cranial imaging. |
| Hickey et al. [18] (1954) | 16,3 | Ireland | Case series | 1/4 | 3 cases with lymphocytic and one with neutrophilic pleocytosis. | MAT | – | 2 cases with concomitant herpes labialis. One patient with meningitis presented with negative MAT during the first symptom week, with posterior positivation. |
| Abdelghani et al. [39] (2021) | 1.5 | Sudan | Prospective cohort | 0/9 | > 70% with neutrophilic pleocytosis | PCR | – | Use of PCR methods in children of the emergency department with pyogenic meningitis diagnosis. 77.8% aged <1 year. Normal CSF in two patients, one with *S. pneumoniae* coinfection, and two with HHV6 coinfection. Average of 2.5 days between onset and lumbar puncture. |
| Romero et al. [40] (2010) | 29,5 | Brazil | Prospective cohort | 0/29 | – | CSF PCR/MAT | – | Without laboratorial data. 29 CSF PCR+, 2 CSF MAT+/PCR-, 5 CSF PCR+/MAT+. 0 positive CSF cultures. |
| Romero et al. [41] (1998) | – | Brazil | Prospective cohort | 0/41 | – | CSF PCR/MAT | – | Without laboratorial data. 39.80% CSF PCR+, 3.88 CSF MAT+, 8.74% CSF ELISA+/MAT+. |
| Torre et al. [42] (1994) | | Italy | Case report | 0/1 | – | CSF MAT | – | Normal CT contrast scan. |
| Silva et al. [43] (1996) | Ranging from 5 to 15 years | Brazil | Prospective cohort | 0/25 | 64% with neutrophil predominance | CSF Serology | – | – |
| Ross et al. [44] (1960) | – | Scotland | Prospective cohort | 0/13 | – | CSF serology/complement-fixation | – | 13/250 tested patients with both serology and complement-fixation positive. |
| Bezerra et al. [45–47] (1993) | – | Brazil | Prospective cohort | 59/18 | – | CSF/MAT | | Normality of physical neurological exam in 89,61%. Meningeal syndrome more common in icteric patients (67,9% *vs.* 32,1%). Abnormal CSF more common in icteric patients (81,3% *vs.* 18,8%). Meningoencephalitis/polyneuritis more common in icteric patients. CSF abnormalities in 95.52%, 88,01% with lymphomononuclear predominance. 74,62% with hyperproteinorrhachia. Positive serology of CSF in 64,18%. Higher CSF glycemic levels. |

*(Continued)*

**Table 2.** (Continued)

| Study | Age | Country | Type | Icteric/ anicteric LAM's | CSF findings | Diagnosis | Interval between fever and meningeal syndrome | Findings |
|---|---|---|---|---|---|---|---|---|
| Gancheva. [23] et al. (2009) | – | Brazil | Case-control study | – | – | MAT/Serology | | 20/94 cases with CNS involvement. 16 with meningitis, 4 with meningoencephalitis. 75% with characteristic neurological examination. Intact cranial nerves. Coma in 20%, seizures in 15%. |
| Panicker et al. [24] (2001) | – | India | Retrospective cohort | – | Lymphomononuclear pattern | Serology | – | Prevalence of various neurological syndromes, but aseptic meningitis/meningoencephalitis predominated (16/40). |
| Coffey et al. [48] (1951) | 27 | United States of America | Case report | 0/1 | Lymphocytic pleocytosis (86%) | MAT | – | – |
| Watkins [49] (1951) | 32 | England | Case report | 0/1 | Lymphocytic pleocytosis (73%) | Serology | 0 days | Evidence of infection in the patient's dog |
| Murgatroyt [50] (1937) | 35 | England | Case report | 1/0 | Neutrophilic pleocytosis (60%). | Xenodiagnosis and MAT | 42 days | The patient had chronic meningitis before an almost complete clinical resolution of Weil's disease. The dark ground illumination of CSF was negative and the diagnosis were made by CSF inoculation in healthy guinea-pigs. Definitive cure of CNS and systemic disease were achieved with anti-leptospiral horse serum administration in CSF and blood. |
| Sakellaridis et al. [15] (2009) | 59 | Greece | Case report | 0/1 | Not mentioned | Serology | – | Meningoencephalitis. MRI scan showed bilateral lateral ventricular enlargement. CSF with increased signal intensity on T1 and decreased on T2. Pathological FLAIR signal. Patient died of acute myocardial infarction. The patient presented a high protein level in CSF, almost equal than the blood protein level. |
| Waggoner et al. [51] (2015) | 25 (mean) | Travellers | Case series | 0/2 | Lymphocytic pleocytosis more common | CSF PCR | – | The two patients had diagnosis with CSF PCR, targeting de 16S rRNA gene.The second patient had a bacterial load in CSF 5- to 10-fold higher than the plasma. His seric serologies were all negative. |
| Middleton [52] (1955) | 42 | England | Case report | 0/1 | Lymphocytic pleocytosis (95%) | MAT | 2 days | Puppy pet examined with evidence of *L.canicola* infection. The patient developed a neuralgic amyotrophy. |
| Marotto et al. [25] (1997) | 4-14 | Brazil | Retrospective cohort | – | Lymphocytic pleocytosis more common (54.4±10.7%) | MAT | – | 23% of cases with meningitis, without neurologic complications or sequelae. |
| Cargill et al. [19] (1947) | – | United States of America | Prospective cohort | 12/1 | Lymphocytic pleocytosis more common | MAT | – | 14 cases of Weil's disease, and 13 with abnormalities in CSF. Only six with clinical signs of meningeal irritation. Xantochromy in 90% of patients (yellow CSF). |

*(Continued)*

| Study | Age | Country | Type | Icteric/anicteric LAM's | CSF findings | Diagnosis | Interval between fever and meningeal syndrome | Findings |
|---|---|---|---|---|---|---|---|---|
| Dittrich et al. [8], (2015) | 25 | Laos | Prospective cohort | – | Mean CSF neutrophils of 30/mm³ and lymphocytes of 21.8/mm³. | CSF PCR | – | 12% of all patients had evidence of leptospiral infection. 23% patients with meningitis, 45% with meningoencephalitis, 7% with acute encephalitis syndrome, and 26 without fulfilling WHO criteria for both meningitis/encephalitis. 3/23 died. |
| Bhatt et al. [53] (2018) | 19 | India | Case report | 0/1 | Lymphocytic pleocytosis | CSF PCR | 2 | MRI revealed multiple hyper intensities in bilateral medial temporal lobes, pons and midbrain suggestive of viral encephalitis. Psychologic sequelae were observed. |
| Puca et al. [11] (2016) | 18 | Albania | Case report | 0/1 | Lymphocytic pleocytosis (100%) | Serology | <4-5 | The patient presented an acute encephalitis as the initial presentation of the disease |
| Bigham [54] (1953) | 25 | United States | Case report | 1/0 | Lymphocytic pleocytosis | MAT | 12 | – |
| Fox et al. [10] (1951) | 11 | Scotland | Case report | 0/1 | – | Complement fixation tests and MAT | <4 | – |
| Mondal et al. [55] (2014) | 32 | India | Case report | 1/0 | Lymphocytic pleocytosis (70%) | Serology | – | Patient presented with a concomitant pancreatitis. |

*Some patients were eliminated, for presenting mild cell count, not configuring an inflammatory CSF. #Interval between symptom onset and lumbar puncture.

Given the scarcity of detailed descriptions of this condition in the literature, a metasummary of reported cases was conducted to delineate the clinical and CSF profile of patients with meningeal syndromes associated with leptospirosis (Fig 2 and Table 2). Overall, a predominantly lymphocytic profile was observed, with mildly increased cellularity, normal to elevated CSF glucose levels, and marked elevation of CSF protein (Fig 2). Nevertheless, substantial heterogeneity in CSF findings was evident, with some cases demonstrating neutrophilic predominance—several of which were confirmed by molecular methods in the CSF [9,18,39,26,32,36,38,51], as well as others showing markedly elevated protein and cell counts [39,32,36,48,49]. These findings underscore the broad spectrum of CSF abnormalities associated with CNS leptospirosis and highlight the diagnostic challenges inherent to this condition.

In milder forms of the disease, it remains unclear to what extent neurological manifestations result from direct pathogen-mediated injury or from host immune responses [1,57,58]. Similar to the systemic infection, CNS involvement is thought to follow a biphasic course [1,58]. There is substantial evidence that, after penetrating host barriers, pathogenic *Leptospira* rapidly disseminates hematogenously, leading to the acute leptospiremic phase [58]. During this stage, the organism can be isolated from multiple tissues, including the CNS, often without eliciting significant inflammatory responses, which explains the absence of pleocytosis or overt inflammatory changes in some cases [58,59]. Clinically, this phenomenon is supported by studies demonstrating molecular evidence of *Leptospira* in the CNS despite unremarkable inflammatory parameters [39]. In contrast, the immune phase of the disease is believed to account for the majority of neurological manifestations, including meningitis, meningoencephalitis, and encephalitis, driven by antigen–antibody–mediated

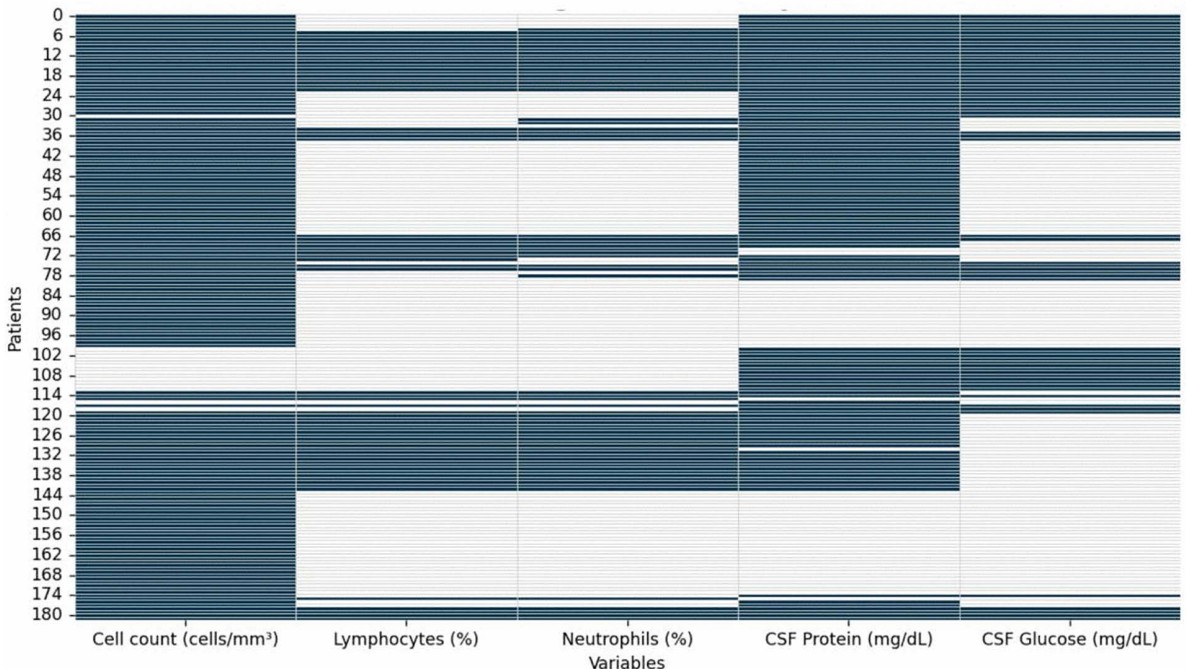

**Fig 1. Heatmap summarizing missing data patterns across CSF parameters from literature and the three reported cases.** The dataset combines variables extracted from published studies reporting CSF findings in leptospirosis-associated meningitis, together with the three additional cases described in the present work. Studies included: Z-Y.

hypersensitivity reactions [59]. Although uncommon, these cases often demonstrate CSF positivity by serological methods such as MAT and ELISA [40–42,44–47].

Herein, we describe three cases of LAM with acute presentation, in accordance with the majority of published reports on LAM. Counterintuitively, the study by Murgatroyd (1937) described a case of chronic LAM with compatible clinical and CSF findings in which *Leptospira* was isolated from CNS by xenodiagnosis [50]. This observation demonstrates that leptospirosis may present with CSF pleocytosis and active central nervous system infection, rather than representing solely an immune-mediated phenomenon [50]. In that case, the patient exhibited immunological anergy to their infecting *Leptospira* strain, requiring experimental administration of equine antileptospiral serum for disease control, suggesting that chronicity resulted from either low bacterial virulence or impaired host immune response [50]. In a convergent manner, Wilson et al. (2014) reported a case of LAM in an adolescent with a four-month history of meningoencephalitis who had a previous diagnosis of severe combined immunodeficiency [27]. This finding supports the hypothesis that persistence of *Leptospira* infection may occur when pathogen clearance after the bacteremic phase is impaired, whether due to host immune deficiency or intrinsic characteristics of the infecting strain. None of our three patients were immunodepressed, and only these two studies reported impairment of immune response among the LAM patients. Thus, isolation of the organism may still occur during the so-called "immune phase." Regardless of complement system integrity, effective immune control of leptospirosis relies on the generation of functional, pathogen-specific antibodies. Host-related factors—including comorbidities, immunosuppression, and genetic polymorphisms—play a critical role in shaping the clinical spectrum of disease [1]. Other studies likewise support the possibility of clinically apparent CNS infection during the bacteremic phase of leptospirosis [18,45].

The pathogenesis of LAM remains poorly understood, largely owing to the frequent underdiagnosis of this condition. Molecular-based prevalence studies conducted in endemic areas generally report higher detection rates, suggesting

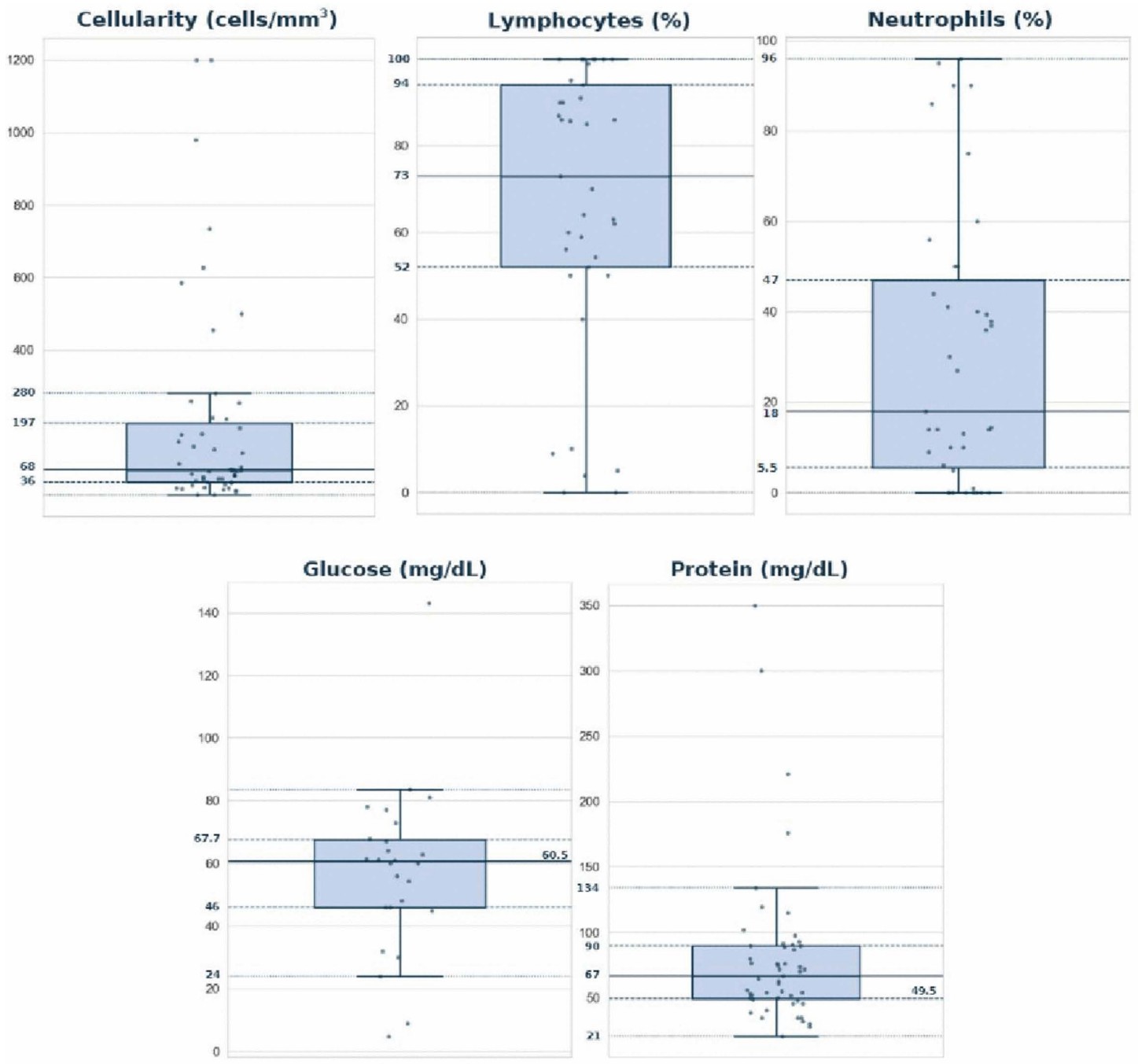

**Fig 2. Boxplot of CSF parameters from literature and the three reported cases.** Extremely high outlier values that distorted the graphical visualization were removed prior to plotting (protein > 400 mg/dL and glucose > 150 mg/dL). Studies included: Z-Y.

that CNS involvement may be more common than clinically recognized [40]. The frequent use of antibiotics in cases of acute meningitis or systemic leptospirosis contributes substantially to this limitation [39,57]. Nevertheless, experimental studies and clinicopathological reports of CNS infection by *Leptospira* spp. provide important insights into the underlying

mechanisms. The immunological nature of LAM is further supported by one study, in which the patient developed meningeal symptoms only after the initiation of antibiotic therapy, consistent with a Jarisch–Herxheimer reaction [38]. This phenomenon, characterized by the release of bacterial cell wall components and subsequent inflammatory cytokine surge, is well described in infections such as leptospirosis and syphilis [60].

Histopathological data remain scarce; however, findings of the studies retrieved in Table 2 reveal cerebral edema and congestion, along with mixed inflammatory infiltrates composed of histiocytes, lymphocytes, and plasma cells [35]. Microglial nodules were also observed across multiple brain regions. Additional findings included perivascular hemorrhage and demyelination, accompanied by varying degrees of lymphohistiocytic infiltration and vascular leakage, consistent with a post-infectious acute disseminated encephalomyelitis–like process without spinal cord involvement. Other studies corroborate these observations, demonstrating predominantly perivascular lymphohistiocytic infiltration within the periarachnoid space [27]. In line with neuropathological findings, LAM is strongly associated with cerebrovascular involvement. Our three cases did not present with CNS bleeding or infarction, but several studies of the meta-summary reported brain vasculopathy (Table 2). Although the exact mechanisms remain unclear, it is widely accepted that immune-mediated vasculitis and/or direct vascular invasion play a central role, contributing not only to intracranial hemorrhage but also to cranial nerve involvement, both of which are frequently observed in this condition [30,45,61]. Additionally, *Leptospira* spp. can interfere with the fibrinolytic system, coagulation pathways, platelet function, and endothelial activation, thereby increasing the risk of hemorrhagic complications [62]. A population-based cohort study from Taiwan demonstrated an increased risk of hemorrhagic stroke among patients with newly diagnosed leptospirosis, even in the absence of overt meningeal involvement [63]. In the reported cases of LAM, cerebrovascular manifestations may include lacunar infarction [13,30,35], subarachnoid hemorrhage [31,45], territorial ischemic stroke [35], Moyamoya-like vasculopathy [8,14], extra-axial hemorrhage [64], and cerebral venous thrombosis [65], among other presentations.

Molecular studies in LAM have expanded understanding of both diagnostic strategies and disease pathophysiology [8,13,27,40,51,66,67]. Evidence suggests that CSF PCR is more sensitive than serological assays, enabling earlier diagnosis and therapeutic intervention [40]. However, its use remains limited in many endemic regions, contributing to the underrecognition of LAM in patients presenting with aseptic meningitis [15,40]. Importantly, the detection of *Leptospira* DNA in patients with LAM does not necessarily indicate active infection, as it may reflect residual genetic material rather than viable organisms capable of replication [15]. These findings suggest that leptospiral aseptic meningitis is likely substantially underreported as a cause of meningitis, particularly in settings where molecular diagnostic tools are not routinely available. Prior studies incorporating PCR-based approaches have demonstrated higher detection rates, reinforcing the notion that the true burden of LAM may be underestimated in the existing literature [66,67].

This study has limitations that should be acknowledged. First, the diagnosis of LAM was not confirmed by direct detection of *Leptospira* in CSF, which may limit diagnostic certainty. However, all cases met accepted diagnostic criteria based on compatible clinical and epidemiological features in conjunction with positive serology, supporting the validity of the diagnoses. Second, the available literature on LAM is scarce and frequently characterized by incomplete reporting and substantial heterogeneity in clinical, laboratory, and diagnostic data, which constrains comparability and synthesis. Third, the retrospective design of this study may be subject to information bias and limits the ability to establish temporal and causal relationships. Finally, it was not possible to further characterize the serovars involved in the studies, as most diagnoses were based on serology/MAT, which cannot reliably identify the infecting serovar [28,66,67].

An estimated one million cases of leptospirosis occur worldwide each year, accounting for approximately 60,000 deaths [1]. Even in its classic clinical presentations, the disease is frequently underrecognized; atypical manifestations further complicate diagnosis and may contribute to substantial underdiagnosis. Treatment of leptospirosis should be initiated based on clinical suspicion, as early antimicrobial therapy is essential regardless of whether neurological involvement results from immune-mediated mechanisms or direct bacterial invasion. Nevertheless, these findings contribute meaningfully to the understanding of CNS involvement in this neglected tropical disease.

## Conclusion

Leptospiral involvement of the CNS remains an underrecognized and likely underreported cause of aseptic meningitis, as reflected by its low frequency in our cohort despite occurring in an endemic setting. The marked variability in CSF findings underscores the diagnostic complexity of this condition and limits reliance on classical laboratory patterns. Differences in detection rates across studies highlight the importance of sensitive molecular techniques, which may reveal a higher true burden of disease. Current evidence suggests a multifactorial pathogenesis, involving both direct bacterial invasion and immune-mediated mechanisms, with host factors influencing clinical expression and disease persistence. From a clinical perspective, LAM should be considered in the differential diagnosis of aseptic meningitis in endemic areas, particularly in the presence of relevant epidemiological exposures and clinical signs of leptospirosis, as early recognition and timely anti-microbial therapy may improve outcomes. From a research standpoint, expanding access to molecular diagnostics and conducting well-designed prospective studies are essential to better characterize disease burden, elucidate underlying mechanisms, and guide evidence-based diagnostic and therapeutic strategies for leptospiral CNS involvement.

## Acknowledgments

The authors gratefully acknowledge the institutional support provided by the Universidade de Fortaleza (UNIFOR), particularly the Postgraduate Program in Collective Health and Medical Sciences, for fostering the development of this research. The authors also thank UNIFOR for its continued commitment to academic excellence and support for clinical and epidemiological research in tropical diseases. We also would like to acknowledge Christus University Center and São José Hospital for Infectious Diseases for their valuable support during the research and data collection.

## Author contributions

**Conceptualization:** Luís Arthur Brasil Gadelha Farias, Osvaldo Mariano Viana Neto, Ednaldo Pereira Lima Sobrinho, Elizabeth de Francesco Daher.

**Data curation:** Luís Arthur Brasil Gadelha Farias, Ednaldo Pereira Lima Sobrinho, Elizabeth de Francesco Daher.

**Formal analysis:** Caroline Lucena de Almeida Vale.

**Funding acquisition:** Geraldo Bezerra Silva Júnior, Elizabeth de Francesco Daher.

**Investigation:** Luís Arthur Brasil Gadelha Farias, Ednaldo Pereira Lima Sobrinho, Caroline Lucena de Almeida Vale, Geraldo Bezerra Silva Júnior, Elizabeth de Francesco Daher, Glaura Fernandes Teixeira de Alcântara.

**Methodology:** Osvaldo Mariano Viana Neto, Ednaldo Pereira Lima Sobrinho, Caroline Lucena de Almeida Vale, Geraldo Bezerra Silva Júnior, Elizabeth de Francesco Daher, Glaura Fernandes Teixeira de Alcântara.

**Project administration:** Luís Arthur Brasil Gadelha Farias, Geraldo Bezerra Silva Júnior, Glaura Fernandes Teixeira de Alcântara.

**Resources:** Caroline Lucena de Almeida Vale, Geraldo Bezerra Silva Júnior, Glaura Fernandes Teixeira de Alcântara, Antônio Silva Lima Neto.

**Software:** Tania Mara Silva Coelho, Maura Salaroli de Oliveira.

**Supervision:** Luís Arthur Brasil Gadelha Farias, Antônio Silva Lima Neto, Maura Salaroli de Oliveira, Lauro Vieira Perdigão Neto.

**Validation:** Tania Mara Silva Coelho.

**Visualization:** Antônio Silva Lima Neto, Tania Mara Silva Coelho, Maura Salaroli de Oliveira, Lauro Vieira Perdigão Neto.

**Writing – original draft:** Luís Arthur Brasil Gadelha Farias, Osvaldo Mariano Viana Neto.

**Writing – review & editing:** Lauro Vieira Perdigão Neto.

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
