## [Decision Letter · Decision Letter 0]

6 Apr 2026

PNTD-D-26-00600

Leptospirosis-Associated Meningitis in an Urban Tropical Endemic Setting in Northeastern Brazil: Three New Cases and a Meta-summary of 176 Reported Cases

Dear Dr. Farias,

Thank you for submitting your manuscript to PLOS Neglected Tropical Diseases. After careful consideration, we feel that it has merit but does not fully meet PLOS Neglected Tropical Diseases's publication criteria as it currently stands. Therefore, we invite you to submit a revised version of the manuscript that addresses the points raised during the review process.

Please submit your revised manuscript within by four weeks. If you will need more time than this to complete your revisions, please reply to this message or contact the journal office at plosntds@plos.org. Please include the following items when submitting your revised manuscript:

We look forward to receiving your revised manuscript.

Kind regards,

Yung-Fu Chang

Academic Editor

Elsio Wunder Jr

Section Editor

Shaden Kamhawi

co-Editor-in-Chief

Paul Brindley

co-Editor-in-Chief

**Additional Editor Comments (if provided):**

**Journal Requirements:**

**Reviewers' Comments:**

Reviewer's Responses to Questions

**Key Review Criteria Required for Acceptance?**

**Methods**

-Are the objectives of the study clearly articulated with a clear testable hypothesis stated?

-Is the study design appropriate to address the stated objectives?

-Is the population clearly described and appropriate for the hypothesis being tested?

-Is the sample size sufficient to ensure adequate power to address the hypothesis being tested?

-Were correct statistical analysis used to support conclusions?

-Are there concerns about ethical or regulatory requirements being met?

Reviewer #1: Methods are OK but clarification/further details should be given to what constitutes lab confirmation of a positive leptospirosis case.

Reviewer #2: Well-structured and methodologically thoughtful, especially the integration of clinical cohort and meta-summary. However, why 3 cases chosen must be answered in the background of high volume literature review cases; why not this is a simple case series with review literature rather making cohort + meta-summary? If it cant be justifiled, better manuscript be revised as simple case series with review literaure.

Definitions are appropriate; consider streamlining for readability by reducing repetition and consolidating overlapping criteria. The literature review methodology would benefit from greater rigor/clarity (e.g., search timeframe, selection process, potential bias). Justification for outlier exclusion thresholds is good; can be strengthened by briefly citing or referencing expected biological ranges.

**Results**

-Does the analysis presented match the analysis plan?

-Are the results clearly and completely presented?

-Are the figures (Tables, Images) of sufficient quality for clarity?

Reviewer #1: Results are appropriate except for the authors' overinterpretation of the infecting serovar from previous publications. Serology CANNOT identify the infecting serovar (multiple publications have proven this) and therefore the table and text need to be revised to reflect this. Additionally, the way that historical publications stated the Leptospira nomenclature is inaccurate. The species should be italicized and the serovar capitalized.

Reviewer #2: Strong descriptive detail and clinically rich cases; consider improving conciseness by focusing on findings most relevant to LAM. Enhance consistency in data reporting (units, structure across cases) to improve readability and comparability. The literature synthesis is valuable; to improve, emphasize key patterns rather than detailed enumeration (e.g., highlight dominant trends over listing serovars). The CSF analysis is a major strength; consider linking findings more explicitly to clinical implications.

**Conclusions**

-Are the conclusions supported by the data presented?

-Are the limitations of analysis clearly described?

-Do the authors discuss how these data can be helpful to advance our understanding of the topic under study?

-Is public health relevance addressed?

Reviewer #1: Conclusions are OK EXCEPT for the assumption that serology can determine the infecting serovar (see above).

Reviewer #2: The conclusion is missing in main text, however in abstract, it is solid but could be sharpened with a clearer clinical or research-oriented closing statement.

**Editorial and Data Presentation Modifications?**

Reviewer #1: (No Response)

Reviewer #2: This is a high-quality, well-researched manuscript with strong novelty in combining clinical data and meta-summary.

Key improvements lie in enhancing clarity, reducing density, and emphasizing your unique contribution and clinical relevance. Tightening the narrative will significantly improve readability and publication impact without altering content depth.

**Summary and General Comments**

Reviewer #1: This is a thorough review of characteristics of aseptic meningitis due to leptospirosis. I think the authors should emphasize strongly that this is likely underreported as a cause of meningitis (the authors have referenced previous studies that looked at PCR diagnosis). The authors also will need to edit the text and table since serology cannot identify the infecting serovar. This was a historic but inaccurate belief, but multiple publications have cautioned that serology cannot identify the infecting serovar.

Reviewer #2: Few points about ABSTRACT: Strong structure and clear integration of cohort + meta-summary; to improve further, clarify the novel contribution more explicitly (e.g., first combined CSF profile synthesis). The results are well presented, but could be strengthened by briefly quantifying key pooled findings (e.g., median CSF values) to increase impact. Conclusion is comprehensive; consider slightly sharpening the clinical takeaway (e.g., when to suspect LAM in practice).

Few remarks on INTRODUCTION: Excellent epidemiological grounding and regional relevance that enhances further by tightening redundancy in pathophysiology description. The knowledge gap is clearly identified; to strengthen it, more explicitly state why prior literature is insufficient (methodological limitations, heterogeneity). The final paragraph is strong but could benefit from a more concise and focused study aim statement.

Similarly DISCUSSION: Highly comprehensive and intellectually strong, particularly in pathophysiology and literature integration. To improve, enhance focus and prioritization, the discussion is dense and would benefit from clearer highlighting of the most important insights. Strengthen direct linkage to your findings (cohort + meta-summary) rather than extended general literature exposition. Some historical and mechanistic details, while interesting, could be condensed to maintain narrative clarity and impact.

The limitations section is appropriate; consider making it slightly more explicit and structured

PLOS authors have the option to publish the peer review history of their article (what does this mean?). If published, this will include your full peer review and any attached files.

Reviewer #1: No

Reviewer #2: **Yes:** prasan kumar panda

**Figure resubmission:**

>While revising your submission, we strongly recommend that you use PLOS’s NAAS tool (https://ngplosjournals.pagemajik.ai/artanalysis) to test your figure files. NAAS can convert your figure files to the TIFF file type and meet basic requirements (such as print size, resolution), or provide you with a report on issues that do not meet our requirements and that NAAS cannot fix. 
---

## [Editor Report · Decision Letter 1]

29 Apr 2026

Dear Farias,

We are pleased to inform you that your manuscript 'Leptospirosis-Associated Meningitis in an Urban Tropical Endemic Setting in Northeastern Brazil: Three New Cases and a Meta-summary of 176 Reported Cases' has been provisionally accepted for publication in PLOS Neglected Tropical Diseases.

Best regards,

Yung-Fu Chang

Academic Editor

Elsio Wunder Jr

Section Editor

Shaden Kamhawi

co-Editor-in-Chief

Paul Brindley

co-Editor-in-Chief

---

## [Editor Report · Acceptance letter]

Dear Mr. Gadelha Farias,

We are delighted to inform you that your manuscript, "Leptospirosis-Associated Meningitis in an Urban Tropical Endemic Setting in Northeastern Brazil: Three New Cases and a Meta-summary of 176 Reported Cases," has been formally accepted for publication in PLOS Neglected Tropical Diseases.

Best regards,

Shaden Kamhawi

co-Editor-in-Chief

Paul Brindley

co-Editor-in-Chief
